# Recent Development of pH-Responsive Polymers for Cancer Nanomedicine

**DOI:** 10.3390/molecules24010004

**Published:** 2018-12-20

**Authors:** Houliang Tang, Weilong Zhao, Jinming Yu, Yang Li, Chao Zhao

**Affiliations:** 1Department of Chemistry, Southern Methodist University, 3215 Daniel Avenue, Dallas, TX 75275, USA; houliangt@smu.edu; 2Global Research IT, Merck & Co., Inc., Boston, MA 02210, USA; weilong.zhao@merck.com; 3Department of Chemical and Biological Engineering, the University of Alabama, Tuscaloosa, AL 35487, USA; jyu56@crimson.ua.edu; 4Boston Children’s Hospital, Harvard Medical School, 300 Longwood Avenue, Boston, MA 02115, USA

**Keywords:** pH responsive polymers, nanomedicine, tumor imaging, drug delivery

## Abstract

Cancer remains a leading cause of death worldwide with more than 10 million new cases every year. Tumor-targeted nanomedicines have shown substantial improvements of the therapeutic index of anticancer agents, addressing the deficiencies of conventional chemotherapy, and have had a tremendous growth over past several decades. Due to the pathophysiological characteristics that almost all tumor tissues have lower pH in comparison to normal healthy tissues, among various tumor-targeted nanomaterials, pH-responsive polymeric materials have been one of the most prevalent approaches for cancer diagnosis and treatment. In this review, we summarized the types of pH-responsive polymers, describing their chemical structures and pH-response mechanisms; we illustrated the structure-property relationships of pH-responsive polymers and introduced the approaches to regulating their pH-responsive behaviors; we also highlighted the most representative applications of pH-responsive polymers in cancer imaging and therapy. This review article aims to provide general guidelines for the rational design of more effective pH-responsive nanomaterials for cancer diagnosis and treatment.

## 1. Introduction

Many diseases originate from abnormal biological processes at the molecular level, such as gene mutation, protein misfolding, and cell malfunctions as a result of infections [1]. Advances in genomics, proteomics, and regenerative medicine have inspired development of numerous therapies and technologies for the treatment of various diseases. With over 10 million new cases every year globally, cancer remains a difficult disease to treat and one of the leading causes for death [2]. Cancer is a prevalent disease that involves a series of genome mutations over time and results in both genetic and phenotypic variations in different tumor cells [3,4]. Targeting cancer-specific biomarkers offers great opportunities for precise tumor detection and efficacious drug delivery, and more importantly, causes minimal side effects on normal cells [5]. However, it is impractical to apply a certain cell surface receptor to a broader range of cancers due to their genetic or phenotypic heterogeneity. In contrast, aerobic glycolysis, also known as the Warburg effect, represents a dysregulated energy metabolism in many types of cancer, where glucose is preferentially taken up and converted into lactic acid [6]. As a result, acidosis has occurred concurrently and evolved as a ubiquitous characteristic of cancer [7]. 

With comparable scales to biologic molecules, nanomaterials have been extensive investigated for biomedical applications over past several decades [8,9]. The nanomaterials are present in small sizes ranging from 1 to 1000 nanometers with correspondingly large surface-to-volume ratio. A multitude of targeting moieties can be incorporated to the surface of nanostructures via diverse engineering procedures [10,11]. The physiochemical properties such as composition, size, and shape can be sophisticatedly tailored for disease diagnosis, treatment, and prevention [12]. Different physiochemical properties like optical, electrical, and magnetic responsiveness have been introduced to nanostructures to facilitate biological diagnostics [13]. One extraordinary property of nanomaterials is that they are generally multi-component systems. Different components or subunits of the nanostructures can be delicately engineered to address the same specific challenge in medicine, resulting in strong cooperative effect that is absent in monomolecular therapeutics [14]. A recent report revealed that over 250 nanomaterial-based technologies or therapies have been approved by the Food and Drug Administration (FDA) or are currently in different stage of clinical trials [15].

There has been explosive development of numerous nanotechnologies to diagnose and treat cancer [16]. The tumor heterogeneity leads to the growing emphasis on customized nanomedicine [17]. Molecular imaging platforms are used to locate tumor and facilitate image-guided surgery [18]. Due to enhanced permeability and retention (EPR) effect, nanoparticles are more likely to accumulate in tumors and spare the surrounding benign tissues [19]. Since abnormal pH has been recognized as a universally diagnostic hallmark of cancer, the design of pH responsive polymers that are capable of altering their chemical or physical properties are of particular interest for cancer theranostics [20,21,22,23]. Moreover, nanoscale polymers with pH responsive segments can strengthen the targeting specificity and hence promote the uptake of nanomedicine into tumor cells [24].

In this perspective, we will review the recent advances in the development of pH responsive nanomaterials for cancer diagnosis and treatment. This article aims to provide general guidelines for the rational design of pH-responsive nanomaterials for the diagnosis and treatment of cancer. Main focus is placed on the polymer designing, mechanistic insights, and specific applications.

## 2. pH-Responsive Polymers

Fabricating pH responsive polymeric nanomaterials has been rapidly developed after the emerging of advanced techniques in polymer synthesis, especially reversible-deactivation radical polymerization such as ATRP and RAFT [25,26]. The technical state of the art allows for precise control over molecular weights and molecular weight distributions, affording polymers with well-defined topology structure [27,28,29]. The pH responsive polymers can be mainly classified into two categories: (1) polymers with ionizable moieties and (2) polymers that contain acid-labile linkages [30]. The former strategy employs a noncovalent transition to achieve pH responsivity. Typical ionizable moieties include amines and carboxylic acids, which can be protonated or deprotonated at different pH values. The alteration of solubility in aqueous medium serves as an amplified signal to represent the change of environmental pH. For the latter, the backbone of polymers usually comprises acid-labile covalent linkages. The decrease of pH is able to trigger the cleavage of these bonds, causing a degradation of polymer chains or a dissociation of polymer aggregates. Compared to polymers with ionizable moieties, polymers containing acid-labile linkages often present a slower internal structural transition due to the nature of covalent bonding, which facilitate their applications in the drug delivery systems.

### 2.1. pH Responsive Polymers with Ionizable Groups 

The pH response arises from the reversible protonation and deprotonation of ionizable groups at the molecular level. The pKa of a polyelectrolyte, which is defined as the pH with equal concentration of the protonated and deprotonated forms, can serve as a critical benchmark to reflect the polymer ionization behaviors at various pH levels [31]. In general, there are two types of ionizable polymers: basic polymers that accept protons at a relatively low pH and acidic polymers that release protons at a relatively high pH. Consequently, these polybases or polyacids can form positively or negatively charged polymer chains at different pH levels. Common basic moieties include amines, pyridines, morpholines, piperazines; and common acidic groups include carboxylic acids, sulfonic acids, phosphoric acids, boronic acids, etc. [30]. Among them, amines, especially tertiary amines, have drawn particular attention due to their ease to prepare and the feasibility to finely tune their pKa [32,33]. It was reported that the amines can present a marginally lower pKa when substituted with longer hydrophobic chains [34]. 

Recently, Gao et al. designed a series of ultra-pH sensitive (UPS) polymers for real-time tumor imaging [35,36,37,38]. These nanoprobes can sharply respond to and amplify in vivo pH signals in a very narrow pH span. The UPS nanoparticles are comprised of an amphiphilic block copolymer: PEG-*b*-PMA, where PEG stands for hydrophilic poly(ethylene glycol) and PMA is a hydrophobic segment based on polymethacrylates with tertiary amine substituent. At physiological pH (7.4), the block copolymers stay as core-shell micelles driven by self-assembly. If the environmental pH is lower than the pKa of the pendent tertiary amines, the amines will get protonated and the micelles will dissociate rapidly into unimers. It is worth noting that the process that comprises assembly and dissociation is completely reversible, and fully determined by ambient pH. Acid-triggered dissociation of micelles enable the increase of fluorescence intensity for molecular imaging and release of cargo for drug delivery.

Yan et al. further expand the UPS polymer design based on biodegradable polypeptides (Figure 1) [39]. They modified the peptides with ionizable tertiary amines for pH responsive behavior. Hydrophilic and hydrophobic blocks were synthesized independently and covalently connected by the copper(I)-catalyzed alkyne-azide cycloaddition (CuAAC) “click” chemistry [40]. The copolymer’s pKa can be readily tuned by altering the ratio of amino substituents. Similar to aforementioned UPS designing, a fluorescent cyanine dye (Cy5.5) was attached at the chain end of the amphiphilic copolymer to convert the subtle pH variation into significant fluorescence intensity change. This study suggested that the ultra-pH sensitivity resulted from the reversible protonation of ionizable amines rather than the peptide backbone. It also inspired the design of degradable ultra-pH sensitive polymers.

Natural macromolecules with ionizable amino acid residues have also been investigated for pH-triggered delivery of imaging and therapeutic agents [41,42,43,44,45]. Engelman and colleagues have been working on the development of novel pH-responsive transmembrane peptides, pH low insertion peptide (pHLIPs), for basic research and translational applications in membrane biophysics and medicine (Figure 2) [46]. These pH-responsive peptides were derived from the C-helix of the protein bacteriorhodopsin [47]. pHLIPs spontaneously self-assemble into a helix structure and insert across the membrane upon exposure to acidic environment [48]. In physiological condition, where the pH is around 7.4, the ionizable acid residues of the pHLIP (red circles) stay negatively charged and the peptide will be weakly bound to the surface of membrane. Once encountering acidic condition like tumor microenvironment, carboxyl group will be protonated and neutralized. Increased lipophilicity (green circles) as a result of ionization drastically enhances the affinity of pHLIP to the hydrophobic inner core of the cellular membrane and triggers the formation of a helix and ensuing insertion across the membrane. When the protonatable carboxyl groups are exposed to the normal intracellular pH, pHLIP gets reversibly deprotonated and anchors in the membrane. The pH-responsive behavior of pHLIP can be easily modified by replacing ionizable aspartic acid residues with glutamic acid residues [43,49] or positively-charged lysine residues [41], or other protonatable amino acids [44].

Polyacids have also been extensively investigated for the design of pH-responsive nanoplatforms. Hyaluronic acid (HA) is a key component of the extracellular matrix and is known to bind to CD44 proteins as a surface receptor on cancer cells [50]. Kono and coworkers reported HA-based pH-sensitive polymer-modified liposomes for tumor-targeted delivery of chemotherapeutics [51]. Instead of simply using HA as targeting moiety, these authors introduced 3-methyl glutarylated (MGlu) units and 2-carboxycyclohexane-1-carboxylated (CHex) units for the design of a new class of pH-responsive polymers with transition pH around 5.4–6.7. Carboxyl group-introduced HA derivatives were prepared by reaction with various dicarboxylic anhydrides. Terminal alkyl chains served as anchor units inserting into the hydrophobic lipid bilayers of liposomes (Figure 3). Upon exposure to acidic endosome pH, both MGlu and CHex modified HA were protonated, enabling lipid membrane disruption and intracellular release of chemotherapeutics. Mechanistic investigation demonstrated that the Mglu/CHex significantly affected the pH-responsive behavior of HA derivatives. Increased hydrophobicity led to a higher transition pH. 

### 2.2. pH Responsive Dissociation Based on Acid-Labile Linkages

Polymers containing acid-labile or base-labile linkages can respond to the change of pH by degradation. Since the tumor microenvironment is slightly acidic compared to normal physiological environment, polymeric nanoparticles containing base-labile linkages have seldom been used for cancer therapy and will not be covered in the perspective. In contrast, polymers with acid-labile linkages have been extensively utilized for the designing of anticancer drug delivery systems [52]. The most commonly used acid-labile linkages are listed in Table 1. The different degradation mechanisms of the linkages and the products after cleavage are described [53,54].

Hydrazone is one of the most explored acid-labile linkages used for acid-responsive dissociation release due to its easy synthesis and moderate sensitivity [55]. Incorporation of hydrazine into the backbone of polymers represents an ideal strategy for the design of tumor-targeted drug delivery systems. Nie group designed a pH-sensitive drug-gold nanoparticle system for tumor chemotherapy and surface enhanced Raman scattering (SERS) imaging (Figure 4) [56]. This multifunctional system comprised of poly(ethylene glycol), doxorubicin (Dox), hydrazone linker, and gold nanoparticles (Au–Dox–PEG). 3-[2-pyridyldithio]propionyl hydrazide (PDPH) was conjugated with Dox in methanol. PDPH acted as a pH-sensitive linker and introduced thiol groups to ensure the anchoring of drug conjugates onto the surface of gold nanoparticles. Gao and coworkers developed a pH-responsive polypeptide–drug nanoparticles for targeted cancer therapy based on well-defined elastin-like polypeptides and acid-labile hydrazone linker [57]. Wang reported a hydrazone-based multifunctional sericin nanoparticles for pH-sensitive subcellular delivery of anti-tumor chemotherapeutics [58].

Acetal and ketal are also commonly used acid-labile linkages, which are very stable under basic conditions, but can be readily hydrolyzed to corresponding carbonyl compound (aldehyde and ketone) and alcohols through a caboxonium ion intermediate upon acidic cleavage [59]. Lu and coworkers reported a novel envelop-like mesoporous silica nanoparticle platform [60]. This system immobilized acetals on the surface of silica (yellow, left) before coupling to gate-keeper nanoparticle (purple, right) (Figure 5). At acidic pH, the acetal was effectively hydrolyzed to remove the gate keepers, allowing the escape of entrapped drug molecules. Liu et al. reported the facile fabrication of acid-sensitive polymersomes for intracellular release of drug over several days [61]. The polymersomes compromising cyclic benzylidene acetals in the hydrophobic bilayers were stable under neutral pH, whereas were hydrolyzed into hydrophilic diol moieties upon exposure to acidic pH milieu. The pH-triggered hydrolysis can be easily monitored by many experimental methods including UV/Vis spectroscopy and TEM. A novel class of acid degradable poly(acetal urethane) was also reported for the construction of acid-degradable micelles for delivery of hydrophobic anti-tumor therapeutics [62]. More recently, Wang group designed a new hyperbranched amphiphilic acetal polymers for pH-sensitive drug delivery [63]. Under neutral conditions, the block copolymers self-assembled into well-defined core-shell micelles. This pH-induced acetal cleavage resulted in the drastic decrease of hydrophobicity and dissociation of micelles. De Geest and coworkers synthesized ketal-containing block copolymers as pH-responsive nanocarriers for the hydrophobic anticancer drug paclitaxel (PTX) [64]. The hydrolysis of the ketal groups in the block copolymer side chains at pH < 5 lead to decomposition of the block copolymer nanoparticles and the PTX release.

Boronate esters, formed between a boronic acid and an alcohol, are stable at neutral or alkaline pH and readily dissociate to boronic acid and alcohol groups in a low-pH environment. Boronate esters have been widely employed to exploit pH-sensitive polymeric carriers for anticancer drug delivery. Messersmith et al. conjugated the boronic acid containing anticancer drug bortezomib (BTZ) to catechol-containing polymers via the boronate ester (Figure 6) [65]. Under neutral or basic condition, BTZ and catechol formed stable conjugates via boronate ester linkers, deactivating the cytotoxicity of BTZ. Upon exposure to lower pH, the conjugates readily release free drugs. Levkin and coworkers reported a dextran-based pH-sensitive nanoparticle system by modifying vicinal diol of dextran with boronate esters [66]. pH-triggered hydrolysis of ester linkers resumed the hydrophilic hydroxyl groups of dextran and destabilized the dug-encapsulated nanoparticles. Kim et al. reported a pH-sensitive nanocomplex by grafting phenylboronic acid (PBA) onto the backbone of poly(maleic anhydride) [67]. Dox and PBA readily formed boronate esters following simple mixing.

## 3. Tunable pH-Responsive Behavior

A fundamental challenge in nanomedicine is the specific delivery the therapeutic or diagnostic agents to the targeted tissues or cells [68,69]. Targeted delivery has shown promise in reducing off-target effect and lowering toxicity [70]. One major consideration in the design of pH-responsive nanomaterials is choosing polymers with pKa values matching the desired pH range. The acidity of gastrointestinal track is drastically different from that of blood [71]. Intracellular compartments such as mitochondria, endosomes, and lysosomes also have slightly different pH values [72]. Even the level of acidosis shows small variation among different types of tumors [6]. Therefore, a key consideration in the design of tumor-targeted pH-responsive polymers is to ensure the polymers are able to differentiate acidic tumor microenvironment from surrounding normal tissues. The variation in the potency of diverse anti-tumor therapeutics also requires tunable release kinetics of payloads.

Reversible protonation of pH responsive polymers leads to the change of the hydrodynamic volume, chain conformation, water solubility and maybe supramolecular self-assembly. Ionization also allows us to modulate the pH-responsive behavior of these polymers. Mechanistic investigation suggests that the acid-base equilibrium in the ionization process is controlled by the balance between hydrophobic interaction and electrostatic repulsion [73]. The transition pH can be tuned by altering the structural factors that affect the hydrophobic or electrostatic interactions. Environmental factors that shift the transition pH such as ionic strength and species are beyond the scope of this review. 

### 3.1. Hydrophobic Modification

Hydrophobic modification is an important approach to modulate the transition pH of ionizable polymers. Incorporating different hydrophobic groups or changing the length of hydrophobic chains can lead to the shift of pKa. Li et al. systemically investigated key structural parameters that affected the transition pH of a series of polymers containing ionizable tertiary amines (Figure 7a) [34]. The hydrophobic interactions can be strengthened by increasing the hydrophobicity of the amine substituents. To accomplish this goal, they synthesized a series of pH sensitive block copolymers with an identical poly(methacrylate) backbone and similar chain length but different linear terminal alkyl groups on the side chain. The polymer with the most hydrophobic pentyl group yielded the lowest pKa at 4.4. Meanwhile, the one with the least hydrophobic isopropyl group as an amine substituent showed the highest pKa, close to 6.6. These results demonstrated that ionizable pH-sensitive copolymers with more hydrophobic amine substituents have a lower pKa. They calculated the octanol–water partition coefficients (LogP) of the repeating unit of hydrophobic segment and used them as a quantitative measure of the strength of hydrophobic interactions. The plot of pKa values as a function of LogP (Figure 7b) showed a linear correlation. To confirm the observation, they synthesized another series of block copolymers with the same backbone and similar chain length, but different cyclic terminal alkyl groups. A plot of transition pH as a function of LogP also showed a similar linear correlation.

To prove the concept that hydrophobic chain length has a significant effect on the pKa value, Li et al. synthesized a series of poly(methacrylate)-poly(ethylene oxide) block copolymers with the fixed hydrophilic poly(ethylene oxide) chains but variable chain lengths of the hydrophobic poly(methacrylate) block (x = 5, 10, 20, 60, and 100), and evaluated the pKa value shift caused by the change of the hydrophobic chain length. Results indicated that the pKa values were inversely proportional to the length of the hydrophobic chain. The transition pH of block copolymers, with the longest hydrophobic segment, yielded the lowest pKa at 6.2 (Figure 7c). In contrast, polymers with the shortest hydrophobic chain length, displayed the highest transition pH around 6.7. The plot of pKa values as a function of the hydrophobic chain length showed a dramatic hydrophobic chain length-dependent transition pH shift.

Hydrophobic modification has also been demonstrated to be a practical strategy in tuning the degradation rate of polymeric systems containing acid-labile linkers. Ramakrishnan et al. investigated how the hydrophobic end-groups affected the degradation of hyperbranched polyacetals (Figure 8) [74]. They changed the structure of hyperbranched polymers by altering the pendant alkyl groups of monomers. Results indicated that the degradation rates of the acid-responsive polymers were significantly affected by the hydrophobic nature the terminal alkyl substituents, where more hydrophobic alkyl groups generally contributed to slower degradation and prolonged release of cargos. 

### 3.2. Copolymerization with Non-Ionizable Polymers

Stayton group reported the development of a series of pH-responsive block copolymers for the delivery of siRNA [75]. These polymers were composed of a positively-charged block of dimethylaminoethyl methacrylate (DMAEMA) to mediate siRNA condensation, and a second endosomal releasing block composed of DMAEMA and propylacrylic acid (PAA) in roughly equimolar ratios, together with butyl methacrylate (BMA). The polymers self-organized into micelles at physiological pH while rendered pH-induced disassembly upon exposure to acidic endosomal pH. The transition pH where reversible micellization occurred could be precisely tuned by systemically changing the fraction of non-ionizable hydrophobic segment (Figure 9), with increased BMA ratio exhibited lower pKa values [76]. Moreover, experimental results also showed that higher fraction of BMA also lead to enhanced homolytic activity, indicating that the transition pH was critical for therapeutic effect associated with pH-triggered drug release. 

### 3.3. Copolymerization with Ionizable Polymers

Incorporation of additional polyelectrolytes that changes both hydrophobic interactions and electrostatic repulsion has also been reported for the modification of the transition pH of polymers. Gao et al. reported a strategy to design ultra-pH sensitive nanoparticles with broad tunability via a copolymerization method [35]. A series of PEO-*b*-P(R1-*r*-R2) block copolymers were prepared. The hydrophobic block P(R1-*r*-R2) contained two randomly distributed ionizable monomers R1 and R2 with different hydrophobicity. The molar fraction of the two monomers (R1 and R2) can be precisely controlled prior to polymerization. A series of ionizable monomers with different alkyl groups (e.g., ethyl, propyl, butyl, and pentyl) were applied in their study. The pKa values of PEO-*b*-PDPA (propyl) and PEO-*b*-PDBA (butyl) were 6.2 and 5.3, respectively. They were able to get a series of PEO-*b*-P(DPA-*r*-DBA) copolymers with pKa values between 6.2 and 5.3 successfully. A plot of pKa values of the obtained block copolymers as a function of the molar fraction of monomers with butyl group yielded a linear correlation. It is worth noting that matching of the hydrophobicity of the two monomers is critical to maintain the ultra-pH sensitivity. Based on this method, they established a library of nanoprobes with pH transitions cover the entire physiologic range of pH (4.0−7.4) (Figure 10). Compared to simple molecular mixture method, this copolymerization strategy achieves robust and broad tunability in transition pH.

### 3.4. Mechanistic Insights into the Tunable pH-Responsive Behavior

Compared to small molecules, polymerization and supramolecular self-assembly represent powerful strategies to produce high-performance materials at nanoscale [77,78,79,80]. Despite the promise in a multitude of biomedical applications, polymeric materials also introduce significantly increased complexity inherent to the multiple interacting components within the systems. The lack of molecular understanding of polymeric systems frequently hampers our ability to rationally design nanomaterials for medicine and healthcare.

In a recent study, Li and coworkers elucidated the molecular pathway of pH-triggered supramolecular self-assembly, which may pave the way for future design of pH-sensitive polymers with easily tunable pKa and pH transition sharpness. They found that hydrophobic phase separation was critical in tuning the pKa values of amphiphilic pH-responsive polymers (Figure 11) [73]. For pH responsive polymers that are not hydrophobic enough at neutral state to self-assemble into nanoparticles, the pH responsive behavior was very similar to that of small molecular pH sensors. The hydrophobic phase separation changes the molecular pathway of protonation in ionizable polymers that can form self-associated nanoparticles. For polyamines, the hydrophobic phase separation significantly shifted the pKa values to lower pH ranges. It is reasonable to hypothesize that the increased hydrophobicity in polyacid systems shifts the transition pH to more basic pH, which is actually validated by hyaluronic acid modified liposomes as described previously [51]. A molecular cooperativity was observed in pH-induced protonation and phase separation, as observed in thermo-responsive polymers and many natural macromolecular systems. This suggests that supramolecular cooperativity may serve as a general strategy in tuning the responsive behavior of stimuli-sensitive nanomaterials [73].

Thayumanavan and coworkers systematically investigated the substituent effects on the pH-sensitivity of acetals and ketals [59]. It is well established that the degradation of acetals and ketals operated through a resonance-stabilized caboxonium ion intermediate [81]. Their study validated that electron-withdrawing substituents generally decreased the hydrolysis rate due to the cationic nature of the intermediate. Benzylidene acetals that were unsubstituted or substituted with electron-withdrawing groups, exhibited drastically slowed hydrolysis under the pH 5 conditions (Figure 12). For examples, half of phenyl-substituted acetal groups were hydrolyzed after 4 min. Incorporating an electron-withdrawing trifluoromethyl moiety increased the half-life to around 12.3 h. The presence of a resonance-based electron-withdrawing functionality further increased the half-life to 51.8 h. The structural fine-tuning of the linkers offers powerful and practical method in future design of acid-degradable polymeric drug delivery systems.

## 4. Applications of pH-Responsive Nanomaterials in Cancer Diagnosis and Treatment

Cancer remains one of the leading causes of death around the world. Nanotechnology has marked a new dimension in fighting against cancer. pH-responsive nanomaterials that target tumor acidosis offer a new paradigm in addressing the deficiencies of conventional chemotherapy in the diagnosis and treatment of various types of cancer.

### 4.1. pH-Sensitive Nanoprobe-Based Fluorescent Imaging and Image-Guided Surgery

Surgery is one of the main options for cancer treatment. When resecting a tumor, a surgeon needs to precisely identify the spread of the cancer. If tumor is not completely removed, there is a great chance of recurrence of cancer. The patients may suffer from crucial organ dysfunction or damage if normal tissues are cut. Thus, intraoperative technologies that can help surgeons visualize the tumor margins during the operations may significantly improve the long-term survival of cancer patients. There is an urgent clinical need for the cheap and practical technology that can universally differentiate tumors from adjacent normal tissues, which could potentially help the surgeons save numerous lives [82,83,84].

Wang et al. reported a pH-responsive nanoparticle-based strategy for the imaging of a broad range of tumors by nonlinear amplification of tumor acidosis signals [85]. As shown in Figure 13, the ionizable amines were neutralized and the amphiphilic block copolymers stayed at micelles state with conjugated fluorescent dyes quenched due a self-quenching Forster resonance energy transfer (HomoFRET) effect. Upon access to acidic tumor microenvironment or internalized into endocytic organelles in the tumor endothelial cells. This system demonstrated a broad tumor specificity with an extraordinary tumor-to-blood ratio (>300-fold) in a variety of tumor models. Following the exceptional imaging outcome, they continued the study in image-guided surgery with several clinical-compatible fluorescent cameras [86]. The real-time tumor-acidosis-guided detection and resection of tumors significantly improved the long-term survival of tumor-bearing mice.

### 4.2. pH-Sensitive Nanoprobe-Based Magnetic Resonance Imaging

Magnetic resonance imaging (MRI) with a contrast agent has been widely used as a powerful tool for cancer diagnosis [87]. However, most MRI contrast agents are non-targeted for cancer and passively distributed throughout the body, which results in a low efficiency and a need for high doses. One approach to addressing these problems is to use pH-responsive amphiphilic block copolymers that would work as platforms for the tumor-targeting delivery of MRI contrast agents and MRI signal enhancement in the tumor region.

Kun Na and coworkers developed a cancer-recognizable MRI contrast agents (CR-CAs) using pH-responsive polymeric micelles [88]. The micelles were self-assembled from copolymers of methoxy poly(ethylene glycol)-*b*-poly(l-histidine) (PEG–p(l-His)) and methoxy poly(ethylene glycol)-*b*-poly(l-lactic acid)–diethylenetriaminopentaacetic acid dianhydride–gadolinium chelate (PEG–p(l-LA)–DTPA-Gd) (Figure 14a–c). In the micelles, p(l-His) blocks were the pH-responsive component. The imidazole groups of p(l-His) blocks were protonated, causing the broken of the micellar structure in acidic tumoral environment. Paramagnetic gadolinium (Gd^3+^) chelates were the MRI contrast agents that enhance the signal intensity upon the exposure to water molecules as the micelles broken. In addition, the CR-CAs’ core was positively charged by protonation of the imidazole groups of p(l-His) blocks after extravasation, which rapidly facilitated accumulation of CR-CAs compared with pH-insensitive micelle-based CAs (Ins-CAs) due to the strengthened interaction between the positively charge CR-CAs and the negatively charged cellular membrane. In vivo, the CR-CAs exhibit highly effective MR contrast enhancement in the CT26 murine tumor region of Balb/c mice, while the MR contrast of the tumor treated with Ins-CAs did not show a significant change over time. CR-CAs enabled the detection of small tumors (3 mm^3^) in vivo within a few minutes (Figure 14d).

Lee et al. encapsulated Fe_3_O_4_ nanoparticles, which are frequently used as a contrast agent for MRI, into the pH-responsive polymeric micelle [89]. This technique has been tested on mice and shown an increased signal intensity over a 24-h period compared to a pH-insensitive contrast agent which did not change in signal intensity.

### 4.3. pH-Responsive Polymeric siRNA Carriers for Cancer Treatment

The use of RNA interference (RNAi) as a tumor-specific gene therapy has attracted increasing attention, and is considered one of the most promising platforms for cancer therapy [90]. However, naked siRNA is unstable, and the effective intracellular delivery of siRNA into the cytoplasm remains a significant challenge. After cellular uptake by passive or receptor-mediated endocytosis, siRNA predominantly locates in endosomes and is degraded by specific enzymes in the lysosome. Thus, to achieve an effective treatment efficiency, escape of siRNA from endosomes to reach cytoplasm is desired. pH-responsive polymeric carriers that facilitate the endosomal escape have been demonstrated one effective approach to mediate intracellular siRNA delivery.

Shi et al. developed several pH-responsive nanoparticle (NP) platforms, captaining Poly(2-(diisopropylamino)ethyl methacrylate) (PDPA) components, for cancer-specific in vivo siRNA delivery [91,92]. PDPA is a type of polycations containing low pKa amines. Low pKa amine group have been shown to exhibit “proton sponge effect” to induce the endosomal escape [93]. Polycations induce the endosomal escape by binding to the oppositely charged cellular membrane and perturbing membrane integrity. In addition, to synergize the endosomal escape, cationic membrane-penetrating oligoarginine grafts or cationic lipid-like grafts were also incorporated into the nanoparticles. After cellular uptake, the rapid protonation of the PDPA segment causes the fast disassembly of the nanoparticles, endosomal swelling, and the exposed membrane-penetrating oligoarginine grafts or cationic lipid-like grafts lead to efficient endosomal escape (Figure 15a). In vivo, the siRNA NPs showed efficient gene silencing and significant inhibition of tumor growth (Figure 15b–d).

Stayton et al. synthesized a diblock polymer composed of PDMAEMA block to condense siRNA and a second block composed of DMAEMA, BMA, and PAA for endosomal-releasing [75]. PAA induced the endosomal escape as a pH responsive membrane-destabilizing polymer, which respond to changes in pH by transitioning from an ionized, hydrophilic structure at physiologic pH (~7.4) to a hydrophobic, membrane-destabilizing conformation at endosomal pH values (<6.6). The diblock copolymers condensed siRNA into 80–250 nm particles. In HeLa cells, the siRNA-mediated knockdown of glyceraldehyde 3-phosphate dehydrogenase (GAPDH) increased as the percentage of BMA in the second block increased.

### 4.4. pH-Responsive Polymeric Anti-Cancer Drug Carriers for Cancer Treatment

Among the various therapeutic strategies for cancer, such as surgery, chemotherapy, radiotherapy, and gene therapy discussed above, chemotherapy is the most often used method in clinical practice [94,95]. However, the side-effects, low therapeutic efficacy and cytotoxicity of the traditional chemical drugs such as Dox, camptothecin and PTX, have hindered the treatment efficiency of cancer chemotherapy [96]. Polymeric nanoparticles have been developed and used as approaches to remove the problems because they can improve pharmacokinetics and biodistribution profiles of anti-cancer drugs via the EPR effect [97]. However, the EPR impact can only improve the accumulation of NPs in tumor tissues, the efficiency of cancer chemotherapy has always still been hindered by the insufficient drug release that induced the concentration of anticancer drugs to the level below the therapeutic window [98]. pH-responsive polymeric drug carriers, which enhance the triggered release of anti-cancer drugs by responding to the tumour acidic microenvironment, have been demonstrated a pathway to address this problem.

For tumor environment triggered drug release, anti-cancer drugs could be ether physically encapsulated into pH-responsive nanoparticles or conjugated to polymer through acid-liable bonds. Kim group physically encapsulated Dox into a variety of pHis-based polymeric micelles for the CT26 tumor treatment. In vitro, the destabilized pH-responsive pHis core enhanced the triggered release of Dox into the cancer cell. In vivo, the Dox-loaded micelles showed a higher CT26 tumour suppression than free Dox did [99]. Etrych et al. conjugated Dox onto an amphiphilic *N*-(2-hydroxypropyl) methacrylamide (HPMA)-based polymer by hydrazone bonds [100]. HPMA copolymer−Dox conjugates were stable in a buffer at pH 7.4, Dox was released in a mild acidic conditions of the tumor microenvironment. In vivo, HPMA copolymer−Dox conjugates significantly reduced the toxic side effects of Dox and enhanced the anti-tumor efficacy.

Liu group reported a drug delivery system with the combination of physical encapsulation and covalent conjugation of anti-cancer drugs for cancer treatment [101]. Dox was conjugated to PEG by Schiff’s base reaction. PEG-Dox prodrug formed stable nanoparticles (PEG-Dox NPs) in water at physiological pH, and encapsulated curcumin (Cur) into the core through hydrophobic interaction (Figure 16a). When the formed nanoparticles, denoted as PEG-Dox-Cur NPs, are internalized by tumor cells, the Schiff’s base linker between PEG and Dox would break in the acidic environment that is often observed in tumors, causing disassembling of the PEG-Dox-Cur NPs and releasing both Dox and Cur into the nuclei and cytoplasma of the tumor cells, respectively. The PEG-Dox-Cur NPs demonstrated enhanced anti-tumor activity in animals compared with free Dox-Cur combination (Figure 16b).

### 4.5. Challenges and Opportunities of Translating pH-Responsive Nanomaterials

Despite the wide range of success evidences from pre-clinical studies, there remains significant challenges for pH-responsive polymeric nanomaterials to enter clinical path. In addition to the specific cases aforementioned, several general aspects should be taken into account. First of all, the formulation of the polymeric nanomaterial applications is much more complex than conventional technology, such as tablets and injections; this brings difficulties in achieving Good Manufacturing Practice (GMP) standards in large-scale production and quality control. Secondly, developing and standardizing biological assays for human toxicity evaluation and monitoring in clinical trials has to adopt the potential novel biology coming from these materials. Besides toxicity, assays are also needed for clinically relevant biomarker identification and validation. In addition, the pharmacokinetics and pharmacodynamics (PK/PD) study is obligated to grasp a deep understanding of the physiochemical properties of polymeric theranostics, adding layers of complexity to high-throughput data acquisition and modeling. From a regulatory perspective, the clear guidelines for the nanomaterials to be developed facing forward translation still requires collaborative efforts.

Promising efforts have been invested towards these challenges. The development of reproducible and stable polymeric nanomaterial batch synthesis protocol can potentially benefit from advanced nanoparticle preparation technology [102,103]. Human organs-on-a-chip platform, mimicking key functions in a human body, offers a great opportunity to address the limited predictive value of conventional pre-clinical cell and animal models, which may be readily applied to polymeric nanomaterials for safety and efficacy evaluations [104].

## 5. Summary and Future Perspective

Various physiological abnormalities, including pH, reactive oxygen species, overexpressed proteins or enzymes, associated with cancer offer great opportunities for the design of stimuli-responsive polymeric materials for tumor-targeted delivery of diagnostic and therapeutic agents. There has been rapid growing interest in the design of pH-responsive polymeric materials to target tumor acidosis, a ubiquitous characteristic shared by almost all types of cancers.

Despite obvious promise in pH-responsive nanomaterials, several challenges may warrant further exploration. First, design of pH-sensitive polymeric materials that are safe for medical and pharmaceutical applications remains a major challenge. It is still highly desirable to explore biodegradable pH-responsive polymers with improved biocompatibility and reduced toxicity. Second, additional efforts should also be directed toward improved tumor targeting efficacy. pH-sensitive polymeric materials that can specifically deliver payload to tumor’s microenvironment with minimal accumulation in normal tissues or endosomal organelles of tumor cells can drastically improve the therapeutic efficacy and decrease the side effects. However, the pH variation between tumor and surrounding normal tissues is not very significant, and different types of tumors may have slightly different level of acidosis. To meet the requirements of various applications that target tumor microenvironment and intracellular organelles pH-responsive polymers with m tunable pKa values are required. Incorporation of multiple modalities into the same polymer or nanoparticle has also shown promise in enhancing tumor targeting efficiency. Intrinsic heterogeneity of tumors may require the design of polymeric nanomaterial system with multiple transition pH values. Moreover, lack of comprehensive mechanistic understanding of pH-triggered responsive behaviors, especially interactions between nanomaterials and in vivo host environment, still hampers our capability in rational design of more effective pH-responsive nanomaterials for cancer treatment. While continuous development of pH-responsive polymers with improved therapeutic efficacy is of great importance, the modification and optimization of existing polymeric materials with validated biocompatibility may represent a more straightforward and efficient strategy.

It’s worth noting that nanomedicine is an interdisciplinary field that requires close collaboration between physicists, chemists, biologists, and physicians. Many reviews’ proof-of-concept studies need a substantial amount of work before potential successful clinical translation. Nevertheless, recent advances in the polymer design, structure-property correlation, mechanistic understanding, and applications of a multitude of pH-sensitive nanomaterials reviewed here provide general guidelines for future rational design of more effective pH-responsive nanomaterials for cancer diagnosis and treatment.

## Figures and Tables

**Figure 1 molecules-24-00004-f001:**
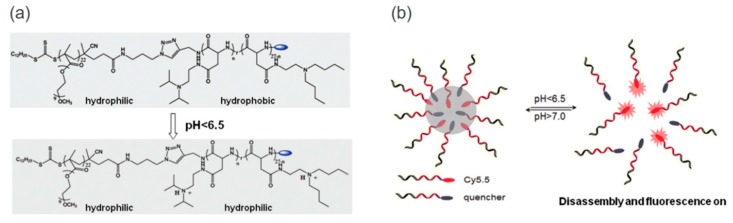
Design of ultra-pH-sensitive polypeptide micelles. (**a**) Structure of amphiphilic copolymers with ionizable tertiary amines and their transition at lower pH. (**b**) pH-triggered disassembly and fluorescence. Reproduced with permission from Liyi Fu, Pan Yuan, Zheng Ruan, Le Liu, Tuanwei Li, and Lifeng Yan, Polymer Chemistry; published by Royal Society of Chemistry, 2017.

**Figure 2 molecules-24-00004-f002:**
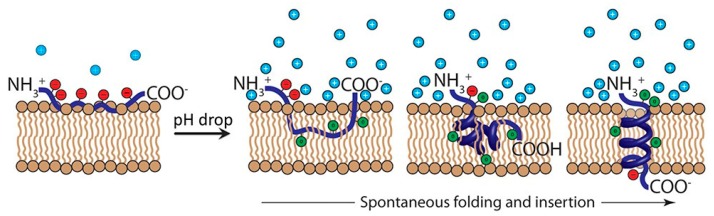
Mechanism of pH (Low) Insertion Peptide (pHLIP) Insertion into the Cellular Membrane. In healthy tissue where the pH is around 7.4, the ionizable residues of the pHLIP (red circles) remain deprotonated and negatively charged, and the peptide resides at or near the hydrophilic surface of the cellular membrane. Upon exposure to acidic tumor microenvironment, the ionizable residues and negatively charged C-terminal carboxyl group of the pHLIP become neutrally charged (green circles). The protonation results in an increase in the hydrophobicity of the pHLIP, triggering the pHLIP to spontaneously fold into a helix and insert across the hydrophobic lipid bilayer of cell membrane, resulting in the formation of a transmembrane helix. Following internalization and exposure to cytosol with pH above 7, C-terminal ionizable residues are deprotonated again and anchor the pHLIP into the membrane. Reproduced with permission from Linden C. Wyatt, Jason S. Lewis, Oleg A. Andreev, Yana K. Reshetnyak, and Donald M. Engelman, Trends in Biotechnology; published by Elsevier, 2017.

**Figure 3 molecules-24-00004-f003:**
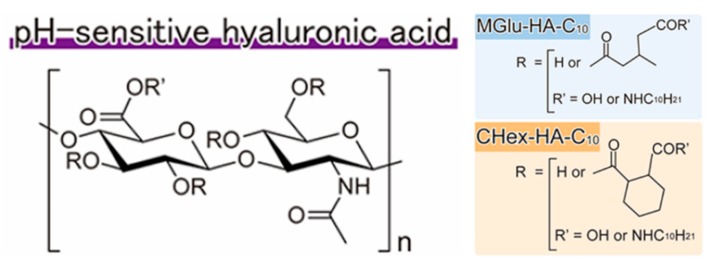
Structures of pH-sensitive modified hyaluronic acid derivatives. Reproduced with permission from Maiko Miyazaki, Eiji Yuba, Hiroshi Hayashi, Atsushi Harada, and Kenji Kono, Bioconjugate Chemistry; published by American Chemical Society, 2018.

**Figure 4 molecules-24-00004-f004:**
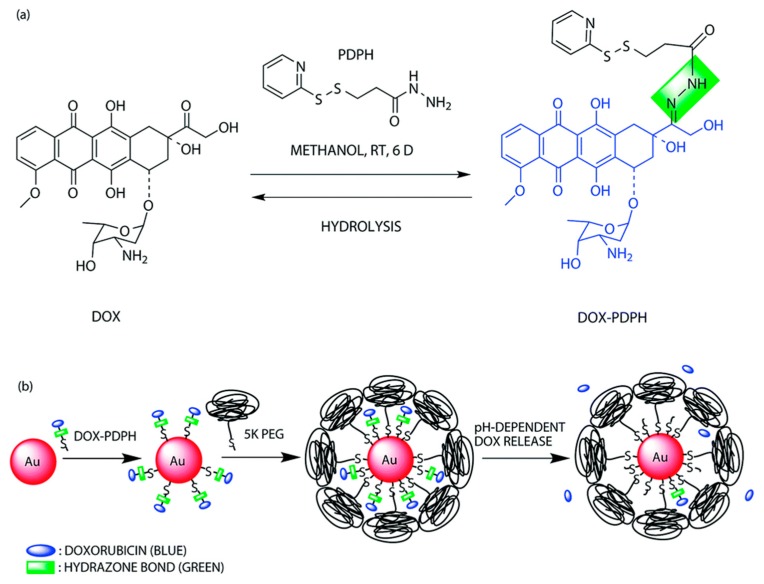
A pH-sensitive drug-gold nanoparticle system. (**a**) Chemical synthesis of the doxorubicin-hydrazone linker conjugate (dox–PDPH). (**b**) Schematic illustration for the synthesis of the multifunctional drug delivery system and its pH-dependent doxorubicin release. Reproduced with permission from Kate Y. J. Lee, Yiqing Wang, and Shuming Nie, RSC Advances; published by Royal Society of Chemistry, 2015.

**Figure 5 molecules-24-00004-f005:**
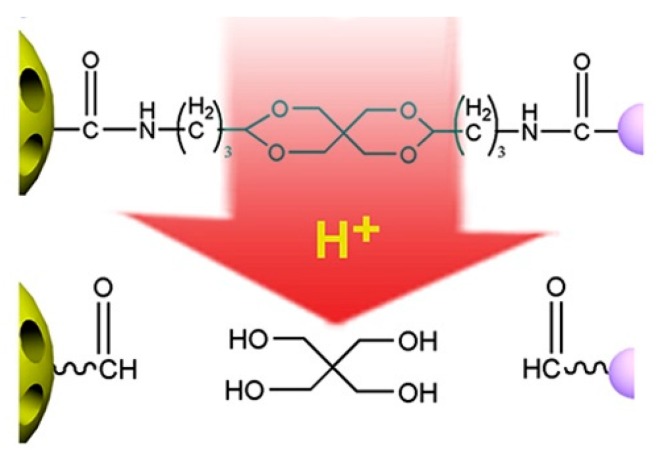
Schematic illustration of the envelope-type mesoporous silica nanoparticle for pH-responsive drug delivery. Reproduced with permission from Yan Chen, Kelong Ai, Jianhua Liu, Guoying Sun, Qi Yin, and Lehui Lu, Biomaterials; published by Elsevier, 2015.

**Figure 6 molecules-24-00004-f006:**
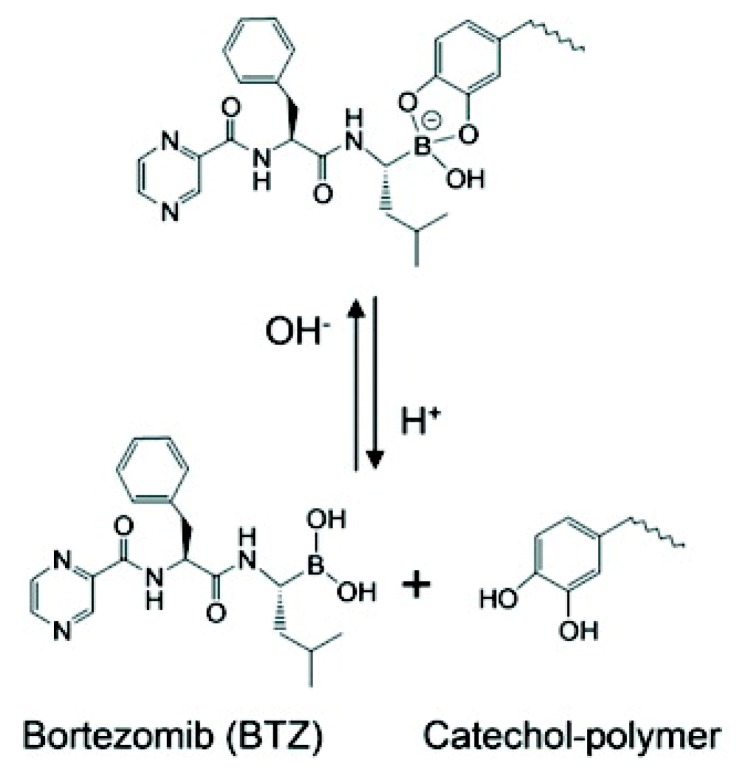
Catechol-containing polymers conjugated the boronic acid containing anticancer drug bortezomib via acid-labile boronate esters. Boronate esters are stable at neutral or alkaline pH and readily dissociate to boronic acid and alcohol groups in acidic environments to release the free active drug. Reproduced with permission from Jing Su, Feng Chen, Vincent L. Cryns, and Phillip B. Messersmith, Journal of the American Chemical Society; published by American Chemical Society, 2011.

**Figure 7 molecules-24-00004-f007:**
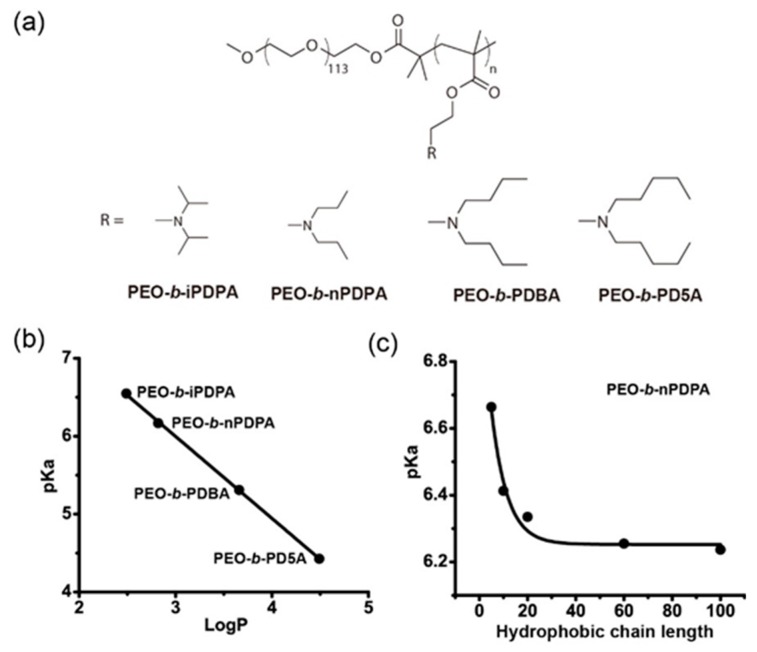
Tunable pKa based on hydrophobic modification. (**a**) Structure of methacrylate-based block copolymers with different alkyl substituents. (**b**) Increase of the hydrophobicity of the alkyl substituents resulted in the linear decrease of pKa. (**c**) Increased hydrophobic chain length lead to decreased pKa in a representative poly(ethylene glycol)-*b*-poly(2-(diisopropylamino)ethyl methacrylate) (PEG-*b*-PDPA) polymer. Reproduced with permission from Yang Li, Zhaohui Wang, Qi Wei, Min Luo, Gang Huang, Baran D. Sumer, and Jinming Gao. Biomaterials; published by Elsevier, 2016.

**Figure 8 molecules-24-00004-f008:**
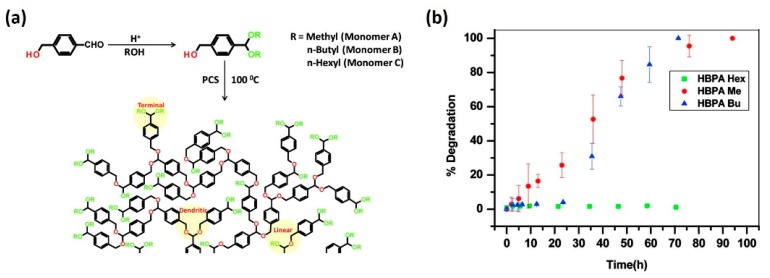
Hydrophobic modification in tuning the degradation rate of hyperbranched polyacetals. (**a**) Structure and synthetic route of hyperbranched polyacetals. Monomers with different hydrophobic substituents (methyl, butyl, and hexyl) were used in the preparation of three model branched polyacetals. (**b**) Branched polyacetal with least hydrophobic methyl groups showed fast degradation within 10 h, whereas polyacetal with most hydrophobic hexyl group showed almost no significant degradation after 4 days. Reproduced with permission from Saptarshi Chatterjee and S. Ramakrishnan. Macromolecules; published by American Chemical Society, 2011.

**Figure 9 molecules-24-00004-f009:**
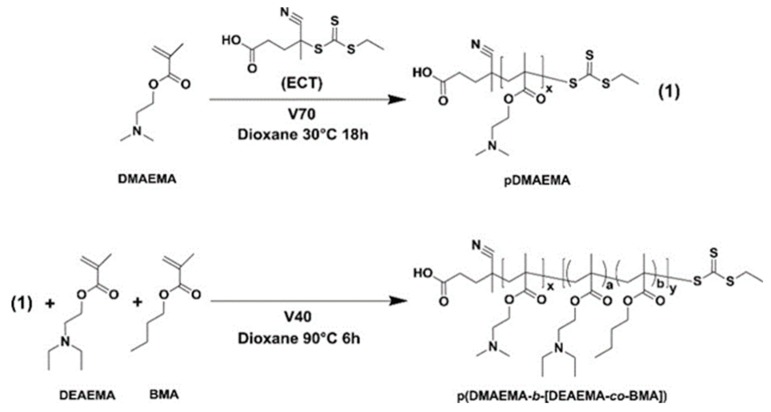
Preparation of a representative copolymer consisting of a cationic poly(DMAEMA) (dimethylaminoethyl methacrylate) block and an endosomolytic hydrophobic block incorporating DEAEMA and butyl methacrylate (BMA) at varying molar feed ratios. Reproduced with permission from Matthew J. Manganiello, Connie Cheng, Anthony J. Convertine, James D. Bryers, and Patrick S. Stayton. Biomaterials; published by Elsevier, 2012.

**Figure 10 molecules-24-00004-f010:**
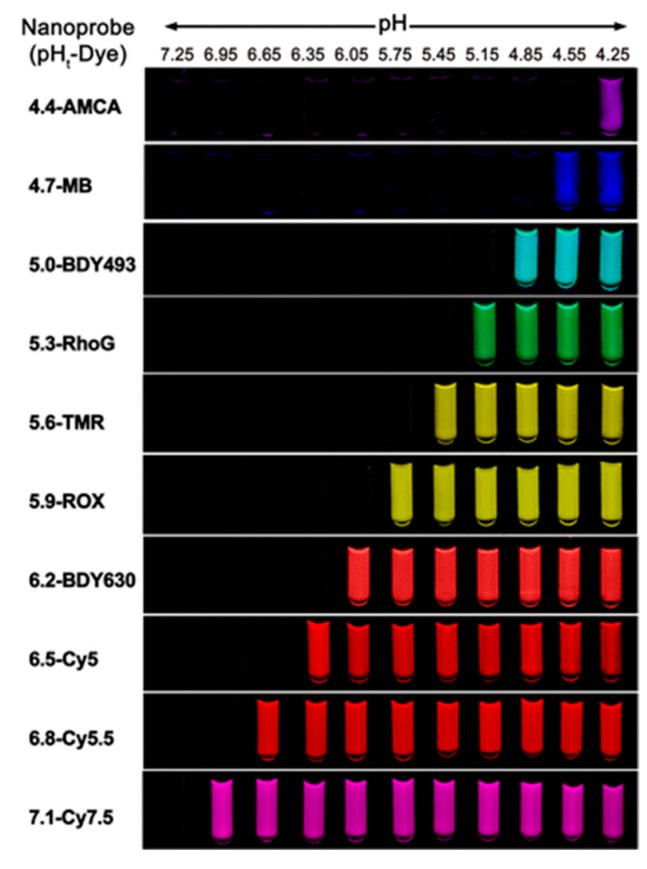
Ultra-pH sensitive (UPS) library based on a copolymerization strategy spanning a wide pH range. Polymers prepared from different monomers ionizable tertiary amines were encoded with different fluorophores. UPS polymers with more hydrophobic repeating unit showed lower transition pH as quantified by fluorescence intensity. Reproduced with permission from Xinpeng Ma, Yiguang Wang, Tian Zhao, Yang Li, Lee-Chun Su, Zhaohui Wang, Gang Huang, Baran D. Sumer, and Jinming Gao. Journal of the American Chemical Society; published by American Chemical Society, 2014.

**Figure 11 molecules-24-00004-f011:**
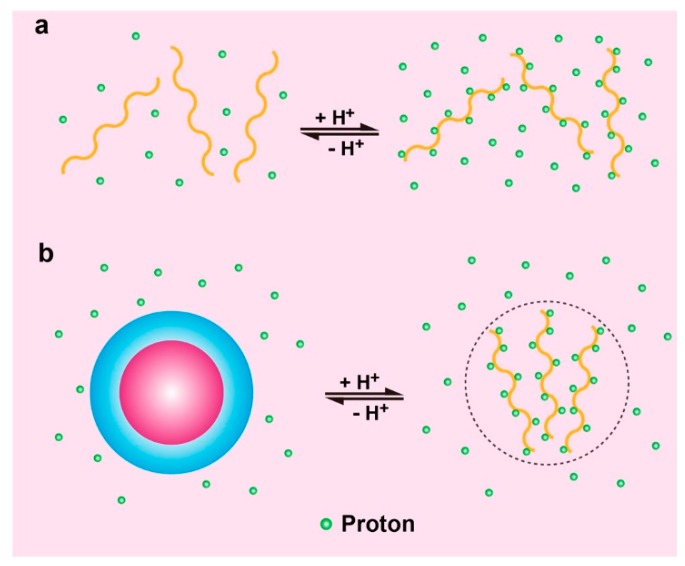
Schematic illustration of phase separation induced cooperativity in pH-triggered supramolecular self-assembly. (**a**) Hydrophilic polymers with ionizable amines were protonated homogeneously, showing no cooperativity. (**b**) Hydrophobic phase separation (e.g., micellization) drove cooperative protonation of amphiphilic pH-responsive polymers. Phase separation also drove the pKa values of polymers to lower pH because of hydrophobic barrier, which required higher proton concentration (lower pH) to initiate the protonation process. Reproduced with permission from Yang Li, Tian Zhao, Chensu Wang, Zhiqiang Lin, Gang Huang, Baran D. Sumer, and Jinming Gao. Nature Communications; published by Nature Publishing Group, 2016.

**Figure 12 molecules-24-00004-f012:**
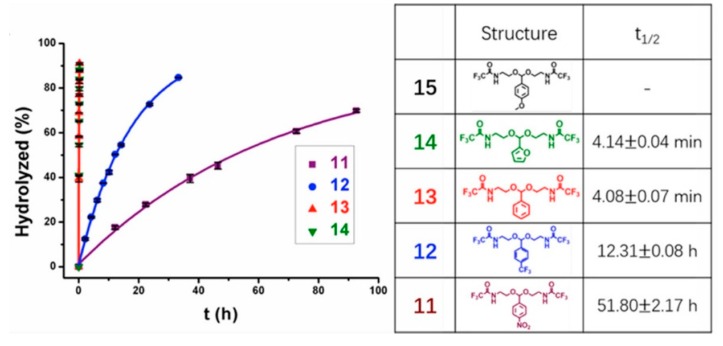
Substituent effects upon the hydrolysis of benzylidene acetals. Electron-withdrawing substituents significantly contributed to decreased hydrolysis rate of acetals. As compared to phenyl substituent with half-life of 4 min, trifluoromethyl increased the half-life of acetal to around 12.3 h. The presence of a resonance-based electron-withdrawing functionality further increased the half-life to 51.8 h. Reproduced with permission from Bin Liu and S. Thayumanavan, Journal of the American Chemical Society; published by American Chemical Society, 2017.

**Figure 13 molecules-24-00004-f013:**
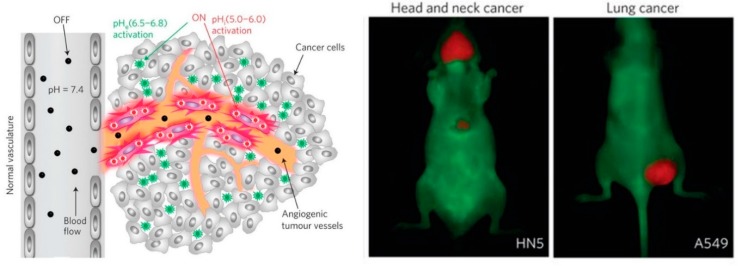
pH-activatable nanoparticles for tumor-specific fluorescent imaging. (Left panel) The nanoparticles stayed in micelle state in normal physiological pH and the fluorescence was quenched as a result of self-quenching Forster resonance energy transfer (HomoFRET) effect. Upon exposure to acidic tumor microenvironment, protonation of tertiary amines lead to the dissociation of micelles and resume of fluorescence. (Right panel) pH-responsive nanoparticles selectively light up tumors instead of surrounding normal tissues in various tumor models. Reproduced with permission from Yiguang Wang, Kejin Zhou, Gang Huang, Christopher Hensley, Xiaonan Huang, Xinpeng Ma, Tian Zhao, Baran D. Sumer, Ralph J. DeBerardinis, and Jinming Gao. Nature Materials; published by Nature Publishing Group, 2014.

**Figure 14 molecules-24-00004-f014:**
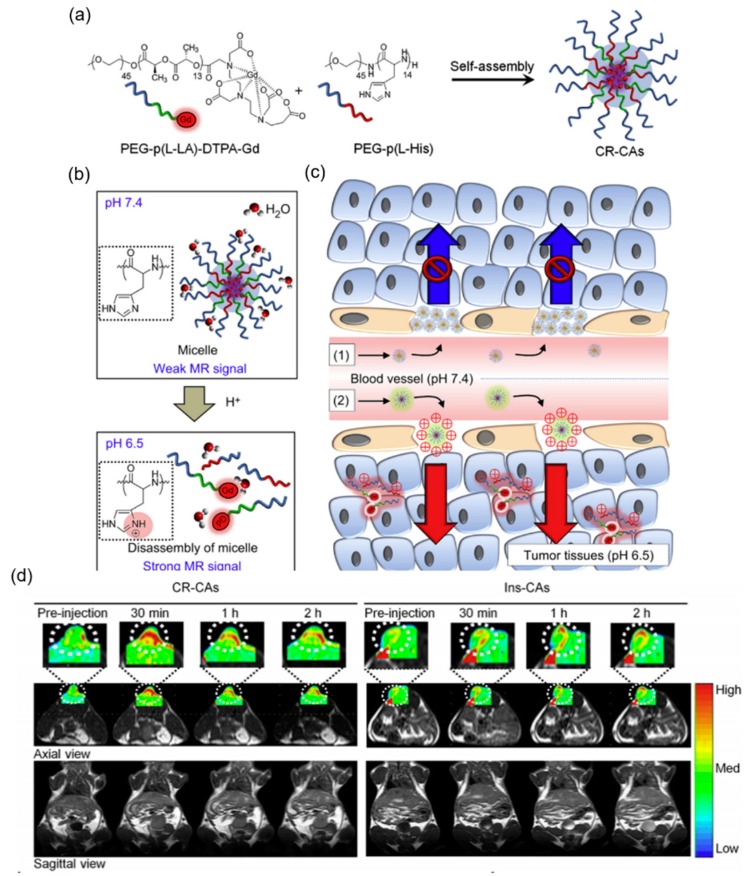
A cancer-recognizable MRI contrast agents (CR-CAs) based on pH-responsive polymeric micelles. (**a**) Schematic representation of the preparation of cancer-recognizable CR-CAs. (**b**) Schematic representation of the pH-dependent structural transformation and related MR signal change in CR-CAs. Inset: Chemical structural representation of the protonation of imidazole groups in PEG-p(l-His) at acidic pH. (**c**) Schematic representation of the tumor-accumulation behavior of (1) conventional micelle-based CAs and (2) CR-CAs. (**d**) Temporal color-coded in vivo longitudinal relaxation time (T1)-weighted MR images of CT26 murine tumor bearing Balb/c mice after the intravenous injection of CR-CAs and pH-insensitive micelle-based CAs (Ins-CAs). Reproduced with permission from Kyoung Sub Kim, Wooram Park, Jun Hu, You Han Bae, and Kun Na. Biomaterials; published by Elsevier, 2014.

**Figure 15 molecules-24-00004-f015:**
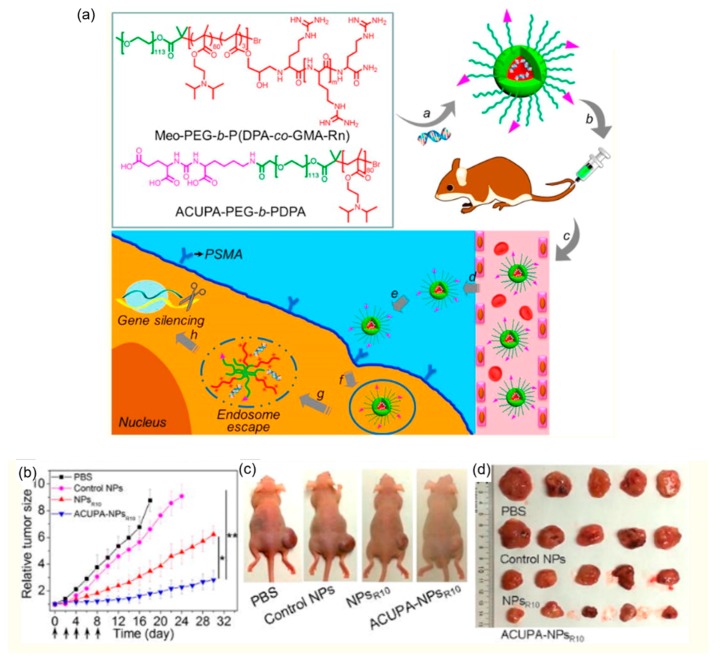
A pH-responsive nanoparticle (NP) platform for cancer-specific in vivo siRNA delivery. (**a**) Molecular structures of the oligoarginine functionalized sharp pH-responsive polymer methoxyl-polyethylene glycol-*b*-poly(2-(diisopropylamino) ethyl methacrylate-*co*-glycidyl methacrylate) (Meo-PEG-*b*-P(DPA-*co*-GMA-Rn, *n* = 6, 8, 10, 20, and 30)) and *S,S*-2-[3–[5–amino-1-carboxypentyl]-ureido]pentanedioic acid functionalized poly(ethylene glycol)-*b*-poly(2-(diisopropylamino)ethyl methacrylate) (ACUPA-PEG-*b*-PDPA) and a schematic illustration of the multifunctional envelope-type NP platform for in vivo siRNA delivery and therapy. (**b**) NPs prepared from Meo-PEG-*b*-P(DPA-*co*-GMA-R10) was denoted as NPs_R10_. Relative tumor size of the Luc-HeLa and PCa cell lines (LNCaP) xenograft tumor-bearing nude mice (*n* = 5) after treatment by Luc siRNA-loaded NPs_R10_ (control NPs) and PHB1 siRNA-loaded NPs_R10_ (NPs_R10_) and siRNA-loaded ACUPA-NPs_R10_ (ACUPA-NPs_R10_). * *p* < 0.05; ** *p* < 0.01. (**c**) Representative photograph of the LNCaP xenograft tumor-bearing nude mice in each group at day 18. (**d**) Photograph of the harvested LNCaP tumors after a 30-day evaluation. Reproduced with permission from Xiaoding Xu, Jun Wu, Yanlan Liu, Phei Er Saw, Wei Tao, Mikyung Yu, Harshal Zope, Michelle Si, Amanda Victorious, Jonathan Rasmussen, Dana Ayyash, Omid C. Farokhzad, and Jinjun Shi. ACS Nano; published by American Chemical Society, 2017.

**Figure 16 molecules-24-00004-f016:**
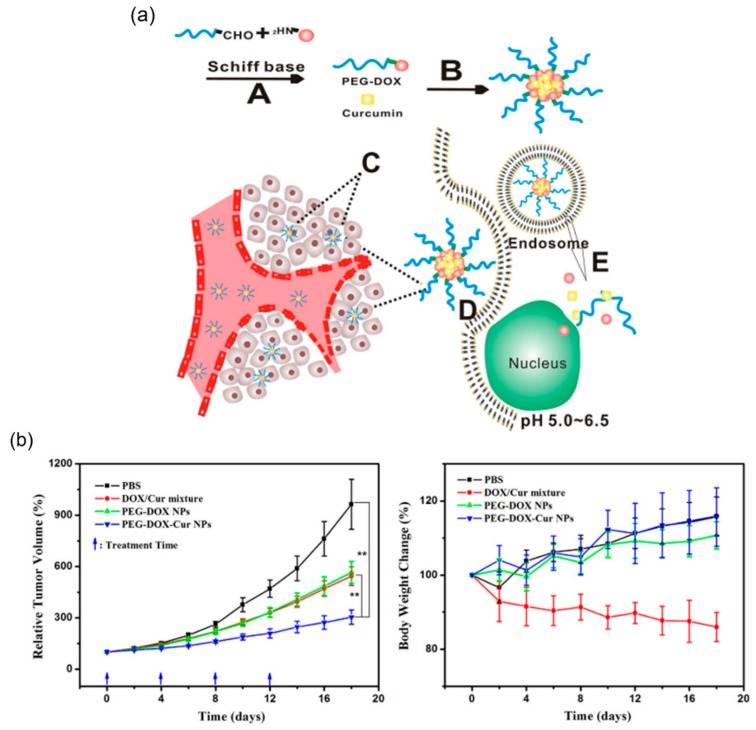
pH-responsive nanoparticles with the combination of physical encapsulation and covalent conjugation of anti-cancer drugs for cancer treatment. (**a**) Schematic illustration of the synthesis and working principle of PEG-Dox-Cur NPs. **A** Synthesis of PEG-Dox NPs via Schiff’s base. **B** Preparation of PEG-Dox-Cur NPs by nanopreciptated technique. **C** Passive tumor targeting was achieved by EPR effect. **D** The PEG-Dox-Cur NPs could be internalized by cancer cells through endocytosis. **E** Dox and Cur were released with the cleavage of the Schiff’s base in tumor cells and diffused into nucleus. (**b**) Relative tumor volume and bodyweight change in Balb/C nude mice after treatment by PBS, free DOX/Cur mixture, PEG-Dox NPs and PEG-Dox-Cur NPs. * *p* < 0.05; ** *p* < 0.01. Reproduced with permission from Yumin Zhang, Cuihong Yang, Weiwei Wang Jinjian Liu, Qiang Liu, Fan Huang, Liping Chu, Honglin Gao, Chen Li, Deling Kong, Qian Liu, and Jianfeng Liu. Scientific Reports; published by Nature Publishing Group, 2016.

**Table 1 molecules-24-00004-t001:** Summary of acid-labile linkages.

Type of Linkages	Structure	Product after Acid Cleavage
Hydrozone	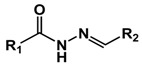	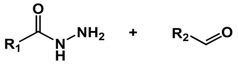
Acetal	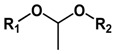	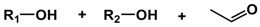
Ketal	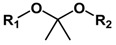	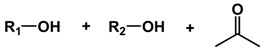
Boronate ester	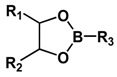	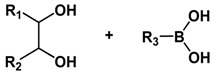

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
