# Peer review of "Recent Development of pH-Responsive Polymers for Cancer Nanomedicine"

_molecules, 2018, doi:10.3390/molecules24010004_

Round 1
Reviewer 1 Report
This review articles describes the recent developments in the pH responsive for cancer nanomedicine.
To make this article stronger , It would be good if authors could add a section describing limitations of pH responsive polymers in terms of clinical translation.
Some of the more specific comments are as follows:
1) Line 29: Biological process can only be at molecular level and not at nanoscale level
2) Line 34: Can you please change there terminology describing cancer as formidable disease?
3) Line 45: The definition of nanometer scale should be between 1-1000 nm rather than 1-100 nm
4) Line 361-363: The sentence formation is messy. Please correct it.
Author Response
To make this article stronger, it would be good if authors could add a section describing limitations of pH responsive polymers in terms of clinical translation.
We appreciate the reviewer’s comment to improve the comprehensiveness of the manuscript. We have added a section (4.5) to discuss current limitations of pH responsive polymers in translating to clinics and potential approaches towards addressing them.
Some of the more specific comments are as follows:
1) Line 29: Biological process can only be at molecular level and not at nanoscale level
Thanks, nanoscale level was deleted from Line 29.
2) Line 34: Can you please change there terminology describing cancer as formidable disease?
Thanks, formidable disease was revised as prevalent disease in Line 34.
3) Line 45: The definition of nanometer scale should be between 1-1000 nm rather than 1-100 nm
Thanks, 1-100 nm was revised as 1-1000 nm in Line 34.
4) Line 361-363: The sentence formation is messy. Please correct it.
Thanks, the sentence in line 361-363 was revised.
Reviewer 2 Report
The authors described recent advances in pH-responsive polymers for responsiveness to tumorours acidic condition. The manuscript is well organized and comprehensive. But, polyzwitterion is also important field in pH-responsive tumor delivery; thus, some of the papers should be added as the references. For example, Ranneh et al. (Angew. Chem. Int. Ed., 57, 5057-5061, 2018) and Mizuhara et al. (Angew. Chem. Int. Ed., 54, 6567-6570, 2018) are quite interesting reports, in which ionizable responses are finely tuned to tumor acidity.
I hope that this manuscript will serve as a great guidance for future design of tumor-resonsive polymers.
Author Response
The authors described recent advances in pH-responsive polymers for responsiveness to tumorours acidic condition. The manuscript is well organized and comprehensive.
Response: We appreciate the reviewer for those very positive comments!
But, polyzwitterion is also important field in pH-responsive tumor delivery; thus, some of the papers should be added as the references. For example, Ranneh et al. (Angew. Chem. Int. Ed., 57, 5057-5061, 2018) and Mizuhara et al. (Angew. Chem. Int. Ed., 54, 6567-6570, 2018) are quite interesting reports, in which ionizable responses are finely tuned to tumor acidity.
Response: Thank you for this constructive comment. We agree that polyzwitterions are promising polymers as pH-responsive nanomaterials for tumor therapy, however their effective transition where polyzwitterions turn into polycations at tumorous pH still falls within the scope of “2.1. pH responsive polymers with ionizable groups”. The above-mentioned two reports are quite interesting, so we have referenced them in line 95.
Reviewer 3 Report
The manuscript entitled “Recent development of pH-responsive polymers for cancer nanomedicine, molecules-402720", prepared by H. T., W. Z., J. Y., Y. L. and C. Z. is a good review with the subject of pH sensitive nanomaterials for cancer therapy.
The subject of this work is good fit to the scope of Molecules, and the research progresses reviewed in the manuscript should demonstrate the benefits of the drug delivery and biomaterials investigations. It seems the authors put several researches and figures together to assemble a review, since they did not give any comments or prospective on each work or each subfield. But basically, in my opinion, there is good novelty, potentially interested for international scientific audiences in this field. Since then, I am suggesting to be accepted by Molecules.
The only question is the cited figures:
1) I cannot find any citation statement for these figures, which makes me believe the figures are made or redrawn by the review authors. At last, I noticed all the figures are cited from the original manuscripts without any modification. So I suggested to make a citation statement for these figures.
2) All the figure content are described perfunctorily. The figure title should give some more detailed information about the related research, and some figures should also be divided as several compartments and give a), b), c) sections and related explanations.
3) Some figures citations and descriptions are not integrate. Since the authors only copied the original pictures but not give some more detailed information or description. Such as figure 13, the A549 is lung cancer cells lines, but the figure definitely showed the xenograft tumor model, so only show the paper here is easy to confuse the audiences. The original paper gave a lot of words on the experiment method and tumor model designing.
Author Response
The only question is the cited figures:
1) I cannot find any citation statement for these figures, which makes me believe the figures are made or redrawn by the review authors. At last, I noticed all the figures are cited from the original manuscripts without any modification. So I suggested to make a citation statement for these figures.
Copyright permissions were provided for all figures.
2) All the figure content are described perfunctorily. The figure title should give some more detailed information about the related research, and some figures should also be divided as several compartments and give a), b), c) sections and related explanations.
All figure contents have been revised to give information that is more detailed. Some of the figures have been divided into several compartments and given a), b), c) sections.
3) Some figures citations and descriptions are not integrate. Since the authors only copied the original pictures but not give some more detailed information or description. Such as figure 13, the A549 is lung cancer cells lines, but the figure definitely showed the xenograft tumor model, so only show the paper here is easy to confuse the audiences. The original paper gave a lot of words on the experiment method and tumor model designing.
All figure contents have been revised to give information that is more detailed.